# Phenotypical, Behavioral, and Systemic Hallmarks in End-Point Mouse Scenarios

**DOI:** 10.3390/ani15040521

**Published:** 2025-02-12

**Authors:** Lidia Castillo-Mariqueo, Daniel Alveal-Mellado, Lydia Giménez-Llort

**Affiliations:** 1Institut de Neurociències, Universitat Autònoma de Barcelona, Cerdanyola del Vallès, 08193 Barcelona, Spain; lcastillo@uct.cl (L.C.-M.); daniel.alveal@autonoma.cat (D.A.-M.); 2Department of Psychiatry and Forensic Medicine, School of Medicine, Universitat Autònoma de Barcelona, Cerdanyola del Vallès, 08193 Barcelona, Spain; 3Departamento de Procesos Terapeúticos, Facultad de Ciencias de la Salud, Universidad Católica de Temuco, Temuco 4780000, Chile

**Keywords:** end-point, animal welfare, euthanasia, bodyweight, sarcopenia, C57BL/6J mice

## Abstract

**Highlights:**

Husbandry homecage behavior and a physical frailty phenotype point to end-point status in mice.Piloerection is the best gross examination hallmark of 16-month-old mice at end-point.Structural kyphosis characterizes end-point mice, while postural kyphosis indicates normal aging.WAT loss and hepatic and splenomegaly indexes can be indirect measures of sarcopenia.Carcass index correlates with piloerection and kyphosis in end-point mice.

**Simple Summary:**

This study examines frailty in 16-month-old C57BL/6J mice, revealing physical signs such as kyphosis and alopecia alongside behavioral indicators like poor emotional responses. Unlike typical end-of-life traits, frail mice exhibited unexpected muscle weight changes, hepatomegaly, and splenomegaly. These findings suggest a complex relationship between physical deterioration and emotional distress in frailty, offering insights into mechanisms relevant to aging populations. Recognizing both physical and behavioral signs in preclinical models is crucial for future research on interventions to address frailty-related decline.

**Abstract:**

The state of frailty is a clinical–biological syndrome that affects the older population with a higher risk of functional dependence. Animal models can provide a tool to study this complex scenario. In the present work, we analyzed the physical and behavioral hallmarks of end-point status in 16-month-old mice (C57BL/6J) according to animal welfare regulations compared to age-matched counterparts with normal aging. A group of 6-month-old mice was added to control for age bias. First, we identified ‘structural kyphosis’ (visible and unmodifiable deformation in locomotion) correlated with piloerection as the hallmarks of the physical frailty phenotype compared to the ‘postural kyphosis’ (adjustment to counteract increased visceral volume but attenuated during locomotion) of old mice with normal aging. Alopecia (barbering) was presented in both old groups. Normal levels of exploratory activity in the corner test for neophobia and triceps surae muscle weight but an increased latency of rearing indicated the poorest emotional phenotype, with a possible contribution of structural kyphosis. The presence of hepatomegaly and splenomegaly counteracted the significant WAT loss commonly associated with end-of-life traits, which should have a normal body weight but preserved muscle mass.

## 1. Introduction

Studies on animal models of aging have critical implications for human aging and age-related diseases [1]. The physical and mental health effects of aging on animals are measurable through gross examination and behavioral assessment [2]. However, there is heterogeneity and complexity in the age-related scenario in old animals, with a reduced survival of animals and a concomitant increase in laboratory costs [3,4,5]. Old mice have a subset of injuries due to the progressive deterioration of organ and system function, which is an indication of frailty and age-related diseases [6,7].

On the other hand, in many studies that use young and adult mice, the scientific end-point is usually related to time, a defined event, or a condition in the animal that occurs or does not occur after a particular intervention [8,9]. When the possibility of animal pain, distress, or suffering exists, researchers should delineate the research objectives and procedures for assessing animal health and ensuring the animal’s well-being [9,10,11]. The end-point at which an animal is euthanized must be established according to clinical or experimental criteria [11,12,13]. Currently, the methods used to assess the condition of a mouse and establish these criteria may include observation of behavior, assessment of physical appearance, and measurement of body weight [13,14,15,16]. Behavioral parameters include observing unprovoked behavior and responses to external stimuli [12,15]. Frequently, physical appearance includes exophthalmia or enophthalmia, runny nose or eye, rough coat, and kyphosis [12]. These findings have been described as standard indicators of ill health, allowing an animal to be monitored over time as its health declines [15,16]. Additionally, decreased food and water consumption is an important sign of declining health [17], which generally results in weight loss [18]. Since a weight loss of 20% alone is a criterion for euthanasia, it is already a reliable predictor of clinical deterioration [19]. Likewise, it has been described that there is an age-related decrease in body temperature [20]. Animals with higher body temperatures and excellent temporal stability tend to live longer, particularly in the C57BL/6 mice strain [20,21]. Although there is consensus that frailty implies multi-organ dysfunction and greater vulnerability to chronic diseases and mortality [7], the systemic effects of aging on mice and its functional implications have not yet been well defined [6,22].

Therefore, the present work aims to determine whether the deterioration of an animal with an end-point indication could be previously inferred not only by hallmark parameters of its physical frailty phenotype (including body weight) but also through ordinary and easy-to-perform homecage housekeeping tasks that allow for the measurement of exploration and neophobia. Since we are interested in targeting body weight loss and the physical frailty phenotype, the liver and spleen were chosen to include the effects of aging at the systemic level. Organometric analysis of post-mortem systemic conditions could indicate that weight can be an indirect measurement of sarcopenia. In addition, we selected the *Triceps surae* muscle as an indicator of sarcopenia and white adipose tissue as an underlying frailty criterion. On the other hand, we also hypothesized that despite presenting positive criteria for the indication of euthanasia in the end-point state, animals with advanced age might maintain their functional performance and body weight regardless of the affectation in the weight of organs or loss of muscular mass. We studied 16-month-old C57BL/6J mice in two clinical aging scenarios: the end-point and normal aging. In addition, a set of 6-month-old mice was added to control for the age factor.

## 2. Materials and Methods

### 2.1. Animals

A total of twenty-two male C57BL/6J mice were used. A group of 16-month-old mice at end-point status according to veterinarian criteria and animal welfare regulations (16M end-point, *n* = 9) was compared with a group of the same age but with normal aging (16M Aging, *n* = 7). The third group, consisting of 6-month-old mice (6M Adult, *n* = 6), was included to monitor the age factor. Animals were housed in three or four per cage and maintained in Macrolon cages (35 × 35 × 25 cm) under standard laboratory conditions of food and water ad libitum, 22 ± 2 °C, a 12 h light-dark cycle starting at 8.00 a.m., and a relative humidity of 50–60%. All procedures adhered to the Spanish legislation on the ‘Protection of Animals Used for Experimental and Other Scientific Purposes’ and the EU Directive (2010/63/UE) on this subject. This study complied with the ARRIVE guidelines developed by the NC3Rs and aimed to reduce the number of animals used [23].

### 2.2. Experimental Design

A cross-sectional study was conducted to assess the appearance of sarcopenia through an indirect study method. The measurements were applied to adult and old animals’ *Triceps surae* muscle compared with old animals that met end-point criteria [20] and those of Talan and Engel [21]: general appearance; skin and fur; eyes; nose, mouth, and head; urine and feces; locomotion. The veterinary staff of the animal house applied the euthanasia criteria.

Before euthanasia, the animals were evaluated with a brief behavioral assessment to verify their physical frailty phenotype, geotaxis, and exploratory activity.

### 2.3. Physical Frailty Phenotype, Geotaxis and Exploratory Activity

The assessment consisted of four consecutive evaluation steps conducted during one hour:(A)Physical frailty phenotype: this includes the body conditions, body weight, alopecia, loss of whiskers, kyphosis, piloerection, tremor, eye discharge, dermatitis, wounds, rectal prolapse, and others. These measurements were made before the exploratory activity and geotaxis. A score of 0 was assigned for normal aspects or 1 for abnormal aspects. A photographic record was taken of each animal to demonstrate these physical aspects.(B)The kyphosis variable was differentiated into ‘postural kyphosis’ and ‘structural kyphosis’ [24]. These were measured during the animals’ locomotion and exploratory activity and later confirmed with the anatomical deformation observed in the postural evaluation.(C)Geotaxis was measured using a 10 × 12 cm grid at a 90° angle. The animal was placed in an inverted position in the grid, and the time it took to reach the vertical position in one trial was measured.(D)Exploratory activity: this was assessed through spontaneous exploration in the corner test. The mice were placed in a 27.5 × 9.5 cm transparent test box and observed for 1 min. The latency to start the movement (taking the hind legs’ motion as a reference), the number of explorations (visited corners), and the latency and number of rearings were recorded. Defecation and urination were also considered.

### 2.4. Hepatic, Splenic, and WAT Indexes Related to Sarcopenia

One hour after the behavioral evaluation, the animals were euthanized (decapitation), and the organs and tissues (liver, spleen, WAT, and triceps surae muscle) were necropsied. The weights of the organs and tissues of each animal were recorded and kept for future analysis.

According to the ‘sarcopenia index’ [25], each animal’s body weight was recorded via the triceps surae muscle’s weight without the calcaneal tendon to calculate sarcopenia in the animals. Second, the sarcopenia index was adjusted by subtracting the larger-volume organs’ weight from carcasses without the liver, spleen, and WAT to control their weight as a confounding bias for ‘bodyweight loss’. Therefore, the differences between the ‘sarcopenia index’ and our new measure, the ‘carcass index’, were verified. We also individually recorded the weight of organs and tissues to demonstrate the differences between the groups.

### 2.5. Statistics

SPSS 15.0 software and the open-source programming language R, version 4.0.3, were used for statistical analysis. Variables that did not present a normal distribution were normalized with a square root or fractionated with fractional rank [26].

The results are expressed as mean ± SEM. ANOVA and Bonferroni post hoc test evaluated differences among three independent groups. To assess the relationship between dichotomous variables (kyphosis, structural kyphosis, and piloerection) and continuous variables (e.g., liver weight, spleen weight, white adipose tissue, rearing latency), a * point-biserial correlation (\ (r_{pb} \)) * analysis was conducted. This statistical method is appropriate when one variable is dichotomous and the other is continuous, allowing us to determine the strength and direction of the association between categorical frailty markers and physiological or behavioral measures. Finally, correlations were analyzed using Pearson’s correlation. Statistical significance was considered at *p* < 0.05.

## 3. Results

The physical frailty phenotype checklist included alopecia, loss of whiskers, kyphosis, piloerection, tremor, eye discharge and swelling, dermatitis and eczema, wounds, and rectal prolapse. The incidence of these variables was indicative of the animals’ end-point status (see Table 1A). The incidence of kyphosis, characteristic of old ages [X^2^, vs. adult mice], was found to be the most sensitive variable in terms of showing the difference between groups [Fisher exact test, per group, with the highest incidence in mice at the end-point [89%, 8/9 mice; Fisher’s exact test (df 2), *p* = 0.002]. In addition, kyphosis was differentiated into two levels of severity: ‘postural’ and ‘structural.’ In end-point animals, a higher prevalence of structural kyphosis was observed, indicating greater severity in this variable [89%, 8/9 mice; Fisher’s exact test (df 2), *p* = 0.001].

The incidence of piloerection was also specific to the end-point [78%, 7/9 mice; Fisher’s exact test (df 2), *p* = 0.004]. Despite being rare, wounds in the body may be present in adults [17%, 1/6 mice] and aging animals [14%, 1/7 mice], but they were more frequent in end-point mice [56%, 5/9 mice]. Eye discharge and swelling [22%, 2/9 mice] and dermatitis/eczema [22%, 2/9 mice] had a low incidence and presented in the end-point group.

As illustrated in Figure 1A, no statistically significant differences were found in the geotaxis, but a tendency to increase the speed to achieve the vertical geotaxis position was observed in old animals [adult, 3.3 ± 0.37 s; aging, 4.1 ± 0.44 s; end-point, 4.7 ± 0.86 s]. Similarly, despite distinctive frailty scores in old animals at the end-point for kyphosis and piloerection, the exploratory test (see Figure 1B) indicated no differences between groups, except those that were age-dependent. An increased fearful response measured as a delay in the latency of the first rearing was seen in old groups compared with adult animals [*p* = 0.042]. As measured by the horizontal and vertical ratios, exploratory activity was also age-dependent [*p* = 0.07], but no differences were observed between either age group (see Figure 1C). The age-dependent trend of changes in emotionality was seen as decreased defecation boli [adult, 1.3 ± 0.61 bolis; aging, 0.4 ± 0.20 bolis; end-point = 0.3 ± 0.17 bolis] and an increased presence of urination [adult, 33.3% (2/6 mice); aging, 14.3% (1/7 mice); end-point, 66.7% (6/9) mice].

Figure 2 illustrates the body weight and different organ and tissue indexes. *Triceps surae* did not show differences between the groups [ANOVA, F (2,19) = 1.475, *p* = 0.25, *n.s.*] but it did show lower weight in end-point animals [adult 30.4 ± 1.21 g, aging 28.8 ± 0.6 g end-point 27.8 ± 1.01 g]. Significant WAT loss was found to be associated with old age [ANOVA, F (2,19) = 8.558, *p* = 0.0022; post hoc 16M end-point vs. 6M adult, *p* = 0.002; 16M aging vs. 6M adult, *p* = 0.026; both old groups vs. adult, *p* = 0.001], [aging vs. end-point *p* = 0.232] but was not enough to differentiate 16M end-point animals from those with normal aging.

Also, this drop in visceral adipose tissue did not translate into body weight loss because the other visceral organs increased with old age and, more specifically, with end-point status. Thus, hepatomegaly [ANOVA, F (2,19) = 9.556, *p* = 0.0013; post hoc 16M end-point vs. 6M adult, *p* = 0.0011] and splenomegaly [ANOVA, F (2,19) = 11.96, *p* = 0.00043, post hoc 16M end-point vs. 6M adult, *p* = 0.0001; both 16M old groups vs. 6M adult, *p* = 0.001] were found. Sarcopenia index [ANOVA, F (2,19) = 1.382, *p* = 0.28, *n.s*.] and its corrected value excluding visceral organs [ANOVA, F (2,19) = 1.247, *p* = 0.28, *n.s*.] did not show group effects.

Table 1B depicts the correlation analysis between piloerection and structural kyphosis, physical frailty makers, and the other weight and behavioral variables. Piloerection correlated with splenomegaly [r^2^ = 0.390 **, *p* = 0.002], WAT loss [piloerection, r^2^= −0.279 **, *p* = 0.004] and hepatomegaly [r^2^ = 0.234 *, *p* = 0.023]. Structural kyphosis and piloerection were positively correlated [r^2^ = 0.274 *, *p* = 0.012]. Structural kyphosis was related to hepatomegaly [r^2^ = 0.245 *, *p* = 0.019] and splenomegaly [r^2^ = 0.184 *, *p* = 0.046] but not to WAT [r^2^ = −0.106, *p* = 0.138]. Only kyphosis seemed to correlate with latency of rearing, albeit it did not reach statistical significance [r^2^ = −0.176, *p* = 0.052].

Figure 3 illustrates the correlation analysis assessing the contribution of the weight of organs (liver and spleen) and tissues (*Triceps surae* and WAT) to body weight. Generally, a negative correlation with the spleen [r^2^ = −0.23 *, *p* = 0.024] and a positive correlation with WAT [r^2^ = 0.449 **, *p* = 0.002] was found. If the body weight adjustment to carcass was carried out (body weight, weight of liver, spleen, and WAT), correlations evidenced, per order of magnitude, WAT [r^2^ = 0.43 ***, *p* = 0.0008], spleen [r^2^ = 0.27 *, *p* = 0.012], liver [r^2^ = 0.22 *, *p* = 0.028], and *Triceps surae* [r^2^ = −0.19 *, *p* = 0.045]. Regarding the correlation between weight and functional performance, a positive correlation in the latency of the first rearing correlated with the absolute [r^2^ = 0.45 *, *p* = 0.048] and adjusted body weight [r^2^ = 0.42, *p* = 0.05] in the 16M end-point animals.

Table 1C shows the correlations between behavioral performance, systemic phenotype, and carcass index. Thus, ‘carcass index’ correlated with rearing latency in the end-point group [r^2^ = 0.42 *, *p* = 0.05]; correlated positively with each of the variables of the systemic phenotype; correlated negatively with liver and spleen [liver r^2^ = −0.22 *, *p* = 0.028; spleen r^2^ = −0.27 *, *p* = 0.012]; and correlated positively with WAT and triceps surae [WAT r^2^ = 0.43 ***, *p* = 0.0008; spleen r^2^ = 0.19 *, *p* = 0.045]. A positive correlation with WAT was observed in the adult group [r^2^ = 0.74 *, *p* = 0.028].

## 4. Discussion

The present work studied 16-month-old C57BL/6J mice who met the euthanization criteria compared to age-matched mice with normal aging and 6-month-old adults. The results showed piloerection and structural kyphosis as their end-point physical frailty hallmarks. In contrast, increased latency of rearing indicated the poorest functional phenotype, which was not justified by muscular loss, since the weight of triceps surae was normal, but could potentially have been derived from structural kyphosis. In addition, hepatomegaly and splenomegaly counteracted the impact that significant WAT loss, commonly associated with age [27], should have on body weight. Therefore, organ indexes were calculated and body weight-adjusted to the ‘carcass index’ to find better indicators of sarcopenia for their end-point status. All of them correlated with piloerection and structural kyphosis.

Aging is characterized by a progressive loss of physiological integrity, leading to impaired function and increased vulnerability to death [28]. In humans, these mechanisms and their triggers have been studied in mice, focusing on frailty [29,30]. Thus, in recent research, frailty has been highlighted as a simple and potentially useful indicator to predict animals’ health status and mortality associated with biomarkers of aging [7]. The end-point status of old animals is defined by a clinical scenario where the severity of their pathologies generates significant functional limitations or compromises the animal’s well-being. Causes of death are often not reported in studies in aged mice [31], and the number of reports is lower when it comes to old animals with naturally occurring pathologies or injuries. In survival studies of this colony, a sex factor has been evidenced with a worse survival rate in females but higher frailty in males [32].

Using our colony of C57BL/6J strain, the gold standard in experimental research [33], the physical frailty phenotype for end-point criteria was similar to that described in previous studies, with differences in the clinical incidence of some physical conditions. In this mouse strain, Pettan-Brewer and Treuting [6] showed that aged mice belonging to the University of Washington colony had four common clinical presentations: rectal prolapse, alopecia and dermatitis, eye lesions, and palpable masses. In addition, dying animals can be euthanized due to relatively nonspecific signs, such as being hunched over, cold to the touch with loss of body condition, and increased respiratory effort [6]. The presence of kyphosis and piloerection stood out as the most frequent reasons, and they were mentioned as nonspecific signs in that study. Background C57BL/6J mice usually present with dermatitis, one of the most commonly observed clinical problems in these mice [12]. Generally, lesions that occur in dermatitis are due to pruritus-induced self-trauma, which progresses from superficial abrasions to deep ulcerations [34]. This situation can explain the wounds in the sample studied in the present work, where they were already present at six months and reached 56% in end-point animals. These ulcerative-like lesions of strain C57BL/6J may be a secondary result of strain-related behavioral characteristics [34].

Piloerection has been described as a sign of dehydration in animals when evaluating their health status [12]. It is also described as an involuntary bristling of the coat as a reflex due to the activation of the sympathetic nervous system [35,36] as part of the evaluation criteria for frailty [36]. At 17 months, males showed more deficits than females in piloerection, indicating more significant signs of frailty in C57BL/6J mice [37]. In addition, these authors found that frailty was linked to pro-inflammatory cytokines in a sex-specific manner.

Furthermore, the high incidence of kyphosis and alopecia exhibited by male mice at 16 months is similar to what was previously reported in this colony [24,29,38,39,40]. Also, in old female mice of this strain but at a higher age, the loss of body mass in senescence has been described and associated with the appearance of other characteristics of the aging phenotype, such as kyphosis, baldness, and loss of coat color [41]. Additionally, the physical frailty phenotype distinguished the type of kyphosis the animals presented with and thus assigned them a severity scale according to their anatomical and functional characteristics [24]. Postural kyphosis refers to compensatory postural adjustments in response to increased visceral volume and disappears or attenuates during locomotion. In the case of ‘structural kyphosis’, an anatomical change is found as an evolution of postural kyphosis with visible and unmodifiable deformation in locomotion. Interestingly, the end-point group showed greater severity in this variable, exhibiting a high prevalence of structural kyphosis. This result could indicate functional alterations that affect the gait and locomotion of the animals, as demonstrated in a previous study of male 3xTg-AD mice, where structural and postural kyphosis constitute a primary impairment that modifies stride and gait speed [24].

Interestingly, exploratory activity and geotaxis showed values that indicated preserved functional levels regardless of age and end-point status. Exploratory activity is associated with behavioral deficiencies in advanced ages, including motor skills [42]. However, certain behavioral domains of the mice are preserved, as is the case with exploratory activity similar to adult mice in Fahlström’s study in females [41]. Also, the corner test for neophobia showed a low latency of movement and showed that the fear of the novelty, expressed as freezing, was almost non-existent in adult animals and slightly increased at 16 months in the aging and end-point groups, as previously reported [5,38,40,43,44].

A progressive reduction in skeletal muscle can result from normal aging without an underlying pathological process, although many chronic diseases can accelerate muscle loss [45,46]. In the present work, body weight and the weight of the triceps surae muscle did not show differences between the groups. However, it was possible to observe a trend towards a decrease in these values to less than 15% body mass in old animals. An element of complexity in this scenario is the heterogeneity of aging. Thus, studies with samples of older animals (males 21–25 months) have shown that the muscle mass of their hind limbs is lower compared to 10-month-old mice, with a significant decrease in daily physical activity and the strength of muscle grip [47]. Therefore, since the 16-month-old animals presented alterations in functional performance related to an increased latency of rearing, we explored whether the loss of muscle mass or kyphosis could explain their poor functional performance. The correlation analysis suggested that structural kyphosis could be related to the reduced latency in performing the first rearing in end-point mice. Their total vertical and horizontal exploration in the corner test did not differ from that exhibited by age-matched counterparts with normal aging.

Geotaxis is a widely used test to measure sensorimotor milestones at the postnatal level in mice [48,49] and is also included in the primary screening of adult animals [50]. It corresponds to an innate response to gravitational signals that require vestibular and motor coordination to orient the body uphill in an angled plane [51]. We have also proposed geotaxis as a functional test to recognize deterioration at this level in old animals [38]. The test requires the constant support of the body by the extremities and a coordinated body balance so that the mouse can rotate its entire body on the declined surface [51]. Although an increase in turning time was detected in old animals, the differences did not reach statistical significance. Moreover, halftime did not exceed 5 s in all groups and was not dependent on body weight, indicating optimal functionality at this sensorimotor level. For comparison, a recent study showed the influence of body weight on the performance of this proof, where at the age of 13 months, overweight animals took about 15 s to complete the test [38].

Liver and spleen organs presented statistically significant differences in the old group with end-point status, showing an increase in the weight of these organs. Previously, in old C57BL/6J mice, an increase in the weight of several systemic organs, namely, the liver, heart, kidney, and spleen, has been shown [52]. These findings also coincide with other strains of mice, such as B6C3F1, where the same phenomenon is observed [53]. The liver has reported weight gain up to 23-28 months, and a decrease is only observed in ancient mice [52]. These organs are relevant in metabolic, inflammatory, and degenerative processes [54]. Thus, spleen weight can indicate alterations in cell number and distribution [55], causing the immune system to be seriously compromised as age increases [54]. Older female C57BL/6J mice have been reported precisely to have an increased spleen [56], and we have also reported splenomegaly to be mainly associated with the female sex and to be exacerbated in 3xTg-AD mice, an animal model of Alzheimer’s disease [57]. Hepatomegaly and liver dysfunction have recently been reported in 16-month-old male and female 3xTg-AD mice at advanced stages of the disease in comparison with their age- and sex-matched non-transgenic counterparts that exhibited liver steatosis [58]. Furthermore, morphological and structural changes due to acute amyloidosis have been detected in the kidney and spleen together with reduced antioxidant defense activity (GPx) in 16-month-old male and female 3xTg-AD mice; all these alterations have been correlated with the neophobia and the anxiety-like behavioral phenotype of this mouse model [59].

In the case of WAT, both the aging and end-point status groups showed a decrease in the weight of this enteric fatty tissue, and this was significant in both cases. Body fat percentage is constant in C57BL/6J mice until 6 months of age; then, it increases between 6 and 12 months [60]. Its decrease in older ages is associated with cachexia, accompanied by decreased body weight [13]. In addition, we detected a negative correlation of this with the spleen and a positive correlation with WAT concerning body weight. Suppose a bodyweight adjustment is made to estimate the weight of the carcass (bodyweight without liver, spleen, and WAT). In that case, these correlations are also evidenced in the liver and *Triceps surae*, which have a negative and positive relationship. These findings coincide with Lessard-Beaudoin’s [52] findings in the liver but differ from the spleen.

The clinical parameters and end-point criteria are essential to determine the point at which the animals will be euthanized, making their use relevant based on their body condition and behavior [8]. In this sense, we have not detected functional impediments or sarcopenia through indirect animal end-point measures. In turn, to diagnose underlying diseases such as vascular, inflammatory, and degenerative conditions, gross in vivo determination of the underlying pathologies of these animals is required, but their confirmation through histological studies carried out after necropsy is also necessary [6].

The limitations of this study include the fact that euthanasia criteria were not applied directly by the researchers but by the veterinary staff at the animal department twice a week, so a study sample was not determined a priori. Only male mice were included because females had higher severity scores according to the criteria of Reynolds et al., 1985 [20], and Talan and Engel, 1986 [21]. In future research, it would be interesting to compare the results of this study with those conducted on females, since their functional profile may differ from that of males.

Functional limitations in gait and exploratory activity may be caused by postural change or by the presence of pain when performing the tests. However, as previously reported with healthy animals, exploratory activity changes, so it may not be painful. Future research may consider changes that predict the end of life, giving researchers a set of criteria for determining when to euthanize older mice.

In conclusion, frailty in 16-month-old C57BL/6J mice is characterized by distinct physical and behavioral changes, including structural kyphosis and emotional distress. These findings enhance our understanding of frailty mechanisms, emphasizing the need for comprehensive assessments in preclinical models to inform interventions targeting frailty-related decline in aging populations.

## 5. Conclusions

This study provides a physical phenotype of findings in male C57BL/6J mice that match end-point criteria within the animal welfare regulations. Thus, it was determined that, despite mice presenting with physical alterations at an age at which they are considered aged animals, optimal geotaxis and exploratory activity are maintained in the animals, similar to their counterparts considered to be aging normally. Moreover, the necropsy of organs and tissues was carried out, and it verified alterations in the weight of the liver and spleen organs, with hepatic and splenomegaly and significant loss of white adipose tissue. However, no alterations at the muscular level in the case of the triceps surae muscle were found, which maintained its weight in adult animals of the same strain. These data expand our understanding of the anatomical changes that occur with aging and provide reference values for further studies in C57BL/6J mice, complementary to those few reported in the literature. These observations also offer the potential to explore the effects of interventions targeting sarcopenia in older mice. Thus, the combination of studies in pathology with in vivo data will fully characterize the effect of proven interventions in multiple chronic diseases and the health of aged mice with a better translation to human aging and age-associated injuries.

## Figures and Tables

**Figure 1 animals-15-00521-f001:**
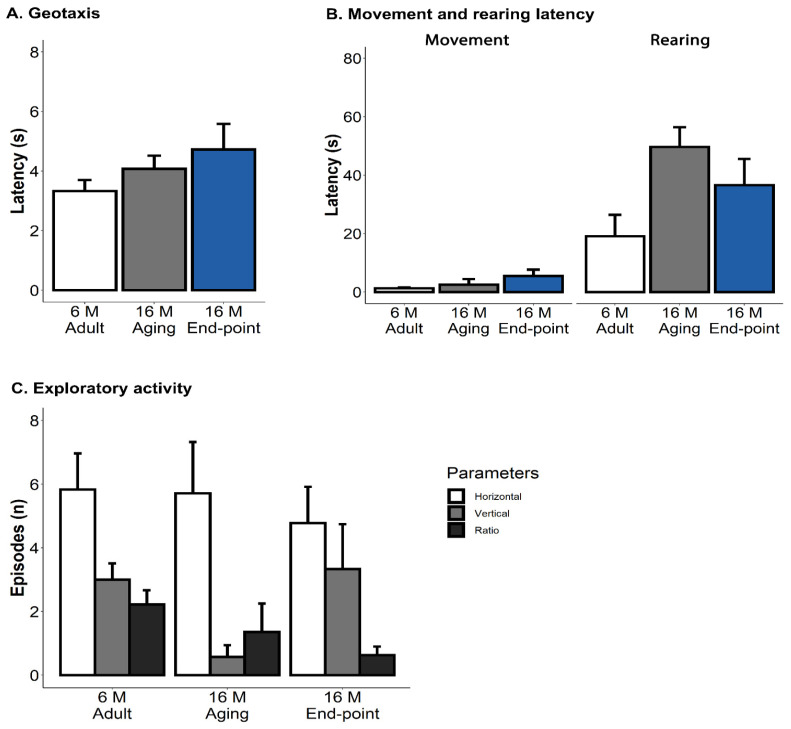
Geotaxis, latency of movement and rearing, and exploratory activity. The results are expressed as mean ± SEM. (**A**) Geotaxis. (**B**) Movement and rearing latency. (**C**) Exploratory activity. Statistics: Student’s *t*-test. adult 6M vs. both old groups aged 16M.

**Figure 2 animals-15-00521-f002:**
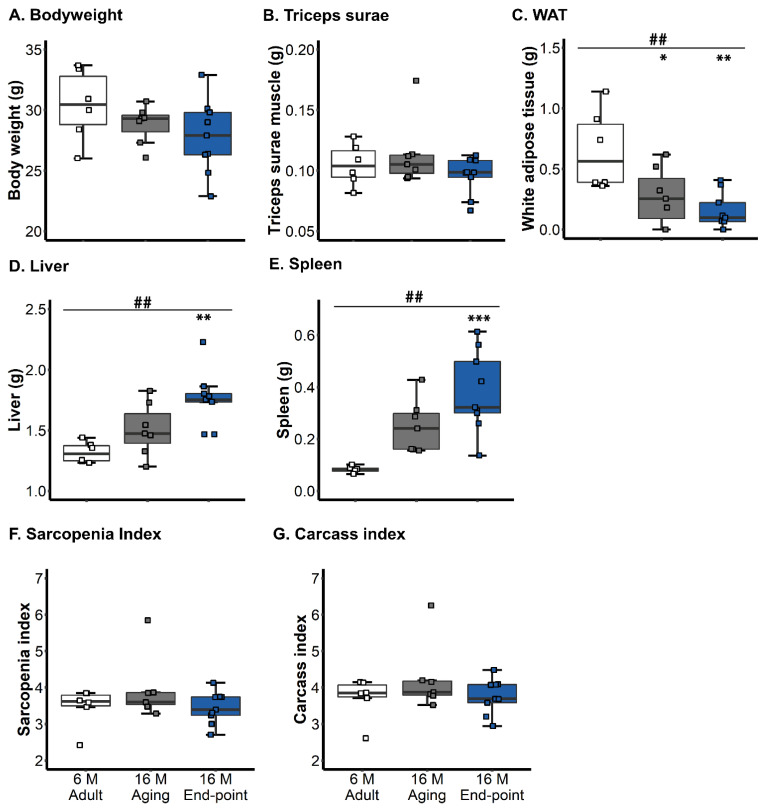
Hepatic, splenic, and WAT indexes related to sarcopenia. Results are expressed as mean ± SEM. (**A**) Body weight; (**B**) *Triceps surae*; (**C**) WAT (**D**) liver; (**E**) spleen; (**F**) sarcopenia index; (**G**) carcass index. Statistics: One-way ANOVA followed by post hoc Bonferroni test, * *p* < 0.05, ** *p* < 0.01 and *** *p* < 0.001. Student’s *t*-test. ^##^ *p* < 0.01, adults 6M vs. both old aged 16M.

**Figure 3 animals-15-00521-f003:**
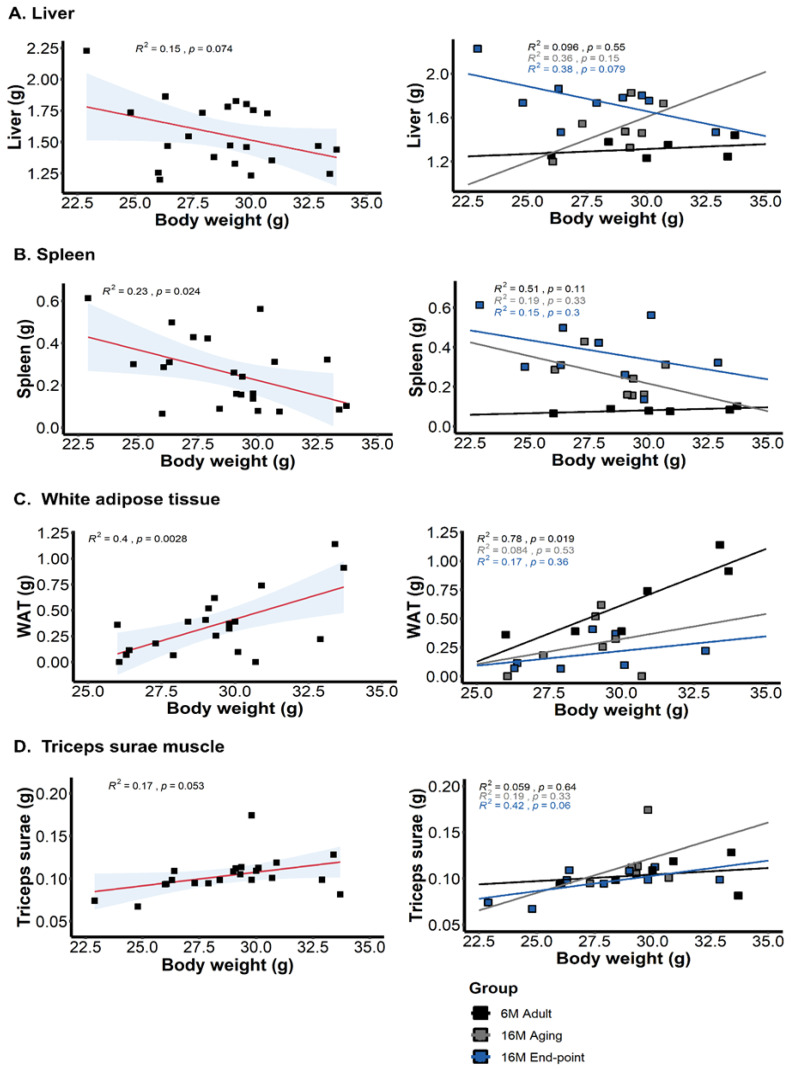
Correlation analysis between body weight and weight of organs (liver and spleen) and tissues (*Triceps surae* and WAT) in C57BL/6J male mice. Meaningful, significant Pearson r correlations between body weight and (**A**) liver, (**B**) spleen, (**C**) white adipose tissue, and (**D**) Triceps surae muscle. Statistics: Pearson r^2^.

**Table 1 animals-15-00521-t001:** Physical frailty phenotype and correlation analysis.

A. **Physical Frailty Phenotype**
**Physical Conditions**	**6M** **Adult (*n* = 6)**	**16M** **Aging (*n* = 7)**	**16M** **End-Point (*n* = 9)**	***p*-Value**
Alopecia	2 (33%)	5 (71%)	4 (44%)	*n.s.*
Loss of whiskers	-	1 (14%)	1 (11%)	*n.s.*
Piloerection	-	1 (14%)	7 (78%)	**, #
Kyphosis	-	3 (43%)	8 (89%)	**, #
Postural	-	2 (29%)	-	*n.s.*
Structural	-	1 (14%)	8 (89%)	**, #
Tremor	-	-	1 (11%)	*n.s.*
Eye discharge/swelling	-	-	2 (22%)	*n.s.*
Dermatitis/eczema	-	-	2 (22%)	*n.s.*
Wounds (face, nose, or periorbital)	1 (17%)	1 (14%)	5 (56%)	*n.s.*
Rectal prolapse	-	-	1 (11%)	*n.s.*
B. **Correlation Analysis Between Hallmarks of Physical Frailty Phenotype at End-Point and Behavioral and Systemic Phenotypes**
**Correlations**	**Kyphosis**	**Structural Kyphosis**	**Piloerection**
**Behaviors**			
90º geotaxis (s)	R^2^ = 0.004 *p* = 0.773	R^2^ = 0.0003 *p* = 0.936	R^2^ = 0.166 *p* = 0.060
Corners (*n*)	R^2^ = (−) 0.089 *p* = 0.177	R^2^ = (−) 0.013 *p* = 0.601	R^2^ = (−) 0.002 *p* = 0.812
Rearing (*n*)	R^2^ = (−) 0.008 *p* = 0.682	R^2^ = 0.002 *p* = 0.811	R^2^ = 0.004 *p* = 0.767
Rearing latency (s)	R^2^ = 0.055 *p* = 0.291	R^2^ = 0.002 *p* = 0.818	R^2^ = 0.131 *p* = 0.098
**Systemic Phenotype**			
Liver (g)	**R^2^ = 0.261 *p* = 0.015 ***	**R^2^ = 0.245 *p* = 0.019 ***	**R^2^ = 0.234 *p* = 0.023 ***
Spleen (g)	R^2^ = 0.137 *p* = 0.089	**R^2^ = 0.184 *p* = 0.046 ***	**R^2^ = 0.390 *p* = 0.002 ****
WAT (g)	R^2^ = (−) 0.176 *p* = 0.052	R^2^ = (−) 0.106 *p* = 0.138	**R^2^ = (−) 0.279 *p* = 0.004 ****
Triceps surae (g)	R^2^ = (−)0.003 *p* = 0.790	R^2^ = (−) 0.132 *p* = 0.096	R^2^ = (−)0.145 *p* = 0.080
**Physical Frailty Genotype**
Structural Kyphosis	-	-	**R^2^ = 0.274 *p* = 0.012 ***
C. **Correlation Analysis Between Carcass Index and Behavioral and Systemic Phenotype**
**Correlations**	**Young (*n* = 6)** **6 Months**	**Aging (*n* = 7)** **16 Months**	**End-Point (*n* = 9)** **16 Months**	**General** ***p*-Value**
**Behavioral Phenotype**
90° geotaxis (s)	R^2^= 0.4 *p* = 0.18	R^2^= 0.13 *p* = 0.43	R^2^= 0.3 *p* = 0.13	R^2^= 0.13 *p* = 0.096
Corners (*n*)	R^2^= 0.28 *p* = 0.28	R^2^= 0.29 *p* = 0.21	R^2^= 0.046 *p* = 0.58	R^2^= 0.0025 *p* = 0.82
Rearing (*n*)	R^2^= 0.011 *p* = 0.84	R^2^= 0.026 *p* = 0.73	R^2^= 0.2 *p* = 0.23	R^2^= 0.069 *p* = 0.24
Rearing latency (s)	R^2^= 0.11 *p* = 0.52	R^2^= 0.027 *p* = 0.72	**R^2^= 0.42 *p* = 0.05 ***	R^2^= 0.16 *p* = 0.066
**Systemic Phenotype**
Liver	R^2^= 0.084 *p* = 0.58	R^2^= 0.27 *p* = 0.23	R^2^= (−)0.44 *p* = 0.051	**R^2^= (−) 0.22 *p* = 0.028 ***
Spleen	R^2^ = 0.51 *p* = 0.11	R^2^ = (−)0.22 *p* = 0.29	R^2^ = (−)0.17 *p* = 0.27	**R^2^ = (−) 0.27 *p* = 0.012 ***
WAT	**R^2^ = 0.74 *p* = 0.028 ***	R^2^ = 0.053 *p* = 0.62	R^2^ = 0.38 *p* = 0.078	**R^2^ = 0.43 *p* = 0.00087 *****
*Triceps surae*	R^2^ = 0.061 *p* = 0.64	R^2^ = 0.22 *p* = 0.29	R^2^ = 0.4 *p* = 0.067	**R^2^ = 0.19 *p* = 0.045 ***
	

Statistics: Table 1A, Fisher’s exact test, ** *p* < 0.01, *n.s.* no significative used for group differences (adult, aging and end-point). X^2^, ^#^ *p* < 0.05 used for age differences (adult vs. old). Table 1B–C, Pearson r correlations test, * *p* ≤ 0.05, ** *p* < 0.01 and *** *p* < 0.001 used for group differences.

## Data Availability

The original contributions presented in this study are included in the article. Further inquiries can be directed to the corresponding authors.

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
